# Restoration of Vegetation Greenness and Possible Changes in Mature Forest Communities in Two Forests Damaged by the Vaia Storm in Northern Italy

**DOI:** 10.3390/plants12061369

**Published:** 2023-03-19

**Authors:** Luca Giupponi, Valeria Leoni, Davide Pedrali, Annamaria Giorgi

**Affiliations:** 1Centre of Applied Studies for the Sustainable Management and Protection of Mountain Areas-CRC Ge.S.Di.Mont., University of Milan, 25048 Edolo, Italy; valeria.leoni@unimi.it (V.L.); davide.pedrali@unimi.it (D.P.); anna.giorgi@unimi.it (A.G.); 2Department of Agricultural and Environmental Sciences-Production, Landscape and Agroenergy-DiSAA, University of Milan, 20133 Milan, Italy

**Keywords:** blowdown forests, Vaia storm, NDVI, secondary plant succession, upward shift in species, vegetation changes, Southern Alps, phytosociology, environmental changes

## Abstract

Windstorms are rare in the Southern Alps, but their frequency is increasing due to climate change. This research analyzed the vegetation of two spruce forests in Camonica Valley (Northern Italy) destroyed by the Vaia storm to evaluate the vegetation responses to blowdown damage. In each study area, the normalized difference vegetation index (NDVI) was used to evaluate the change in plant cover and greenness from 2018 (before the Vaia storm) to 2021. Furthermore, floristic-vegetation data were analyzed to identify current plant communities and develop models of plant succession. The results showed that the two areas, although located in different altitudinal vegetation belts, are undergoing the same ecological processes. NDVI is increasing in both areas, and pre-disturbance values (~0.8) should be reached in less than ten years. Nevertheless, the spontaneous restoration of pre-disturbance forest communities (*Calamagrostio arundinaceae-Piceetum*) should not occur in both study areas. In fact, the two plant succession trends are characterized by pioneer and intermediate stages with young trees of *Quercus petraea* and *Abies alba*, typical of more thermophilic mature forest communities compared to pre-disturbance ones. These results could reinforce the trend of the upward shift in forest plant species and plant communities in response to environmental changes in mountain areas.

## 1. Introduction

Severe storms are increasingly recurrent extreme events due to climate change [1,2,3]. These phenomena, more and more frequent also in the Alps [1,4], cause significant damage and imbalances to ecosystems, including forests [5]. Windthrow and windsnap represent the two main types of forest damage that are observed immediately after a strong windstorm. Such damage has long-term repercussions on the economic aspects (loss of forest productivity), civil security (stability of trees and slopes) [6], and landscape and biodiversity (changes in the biotic and abiotic components of the ecosystems) [7,8].

Blowdown (windthrow and windsnap) generally affects only one part of the forest plant community: the trees. Sometimes, it also affects taller shrubs, while small shrubs and herbaceous plants rarely suffer direct wind damage but can be damaged from downed trees or from their uprooting. The plant communities present in a forest after a severe storm can therefore become forest communities without standing trees, and, thus, without that part of the plant community that most depends on human management. In fact, those who manage the forests rarely act on shrubs and, even less, on herbs, except in cases such as ecological restoration, soil bioengineering, or nature conservation projects [9]. Furthermore, the soil and the seed bank are maintained after the blowdown event. This differentiates windstorm damage from that of other more destructive phenomena (e.g., landslides, avalanches, floods, and volcanic eruptions) and plays a crucial role in forest regeneration capacity and in the diversity of plant communities [10].

The plant communities of the forests destroyed by the wind (mostly shrubs and herbaceous plants) are the first stage of a secondary succession that, during tens of years and after various succession stages, will culminate in plant communities being in equilibrium with environmental factors (the anthropogenic factor included here). Many environmental factors are changing abruptly (such as rising temperature and carbon dioxide, and increasing transient disturbances, including wildfire, drought, windthrow, biotic attack, and land-use change) and forest dynamics with them [5]. For this reason, the final stage of the succession that occurs after the blowdown disturbance could be: (1) a forest similar to the pre-disturbance one, (2) a different forest (from a floristic, physiognomic and/or ecological point of view) compared to the pre-disturbance one, or (3) another type of vegetation (shrubland, grassland etc.). The study of spontaneous vegetation succession (and the hypothetical mature stage) could be of considerable help for territorial managers. In fact, such studies would allow finding the best solutions to restore the areas damaged by the storms and accelerate, as far as possible, the successions and the regeneration of the forests [11,12].

In the Southern Alps, there are no studies on the plant succession of forests destroyed by wind because in this geographical area, damage from severe storms is very rare compared to what occurs in Northern and Central Europe [4,13]. In fact, the Alps act as a barrier to storms that develop at higher latitudes [13]. Despite this, the Italian Alps in the last 40 years have been affected by two strong storms (Vivian in 1990 and Vaia in 2018) that destroyed several hectares of forest. While the Vivian storm affected North-Western Italy only marginally, the effects of the Vaia storm were more intense and extensive [4,14]. The Vaia storm hit the North-Eastern part of Italy from 28 to 30 October 2018, with winds exceeding 200 km h^−1^ and strong rainfall (more than 350–400 mm). It caused extensive forest damage in 494 Italian municipalities [14]. Vaia destroyed or severely damaged forests in about 42.500 ha. Most of them were Norway spruce (*Picea abies* (L.) H. Karst.) forests, with an estimated stock of fallen trees of 8.5 million of cubic meters [14,15].

Lombardy was one of the regions of Northern Italy mostly affected by Vaia. More than 2200 ha of forest were destroyed and over 70% of the damage involved spruce forests [16]. Many of the spruce forests of this Region belong to the *Calamagrostio arundinaceae-Piceetum*, one of the most widespread plant associations in Lombardy from 1100 m (mountain vegetation belt) to 1800 m a.s.l. (high mountain belt) [17]. Given that storms such as Vaia are likely to increase more and more in the future due to climate change [1,2,5], vegetation responses to these disturbances should be studied in the Southern Alps, as has already been conducted in the Swiss Alps [18,19], in order to improve the management practices of damaged forests and reduce their vulnerability [20]. Furthermore, considering Lombardy forests of *Calamagrostio arundinaceae-Piceetum* damaged by the storm, it would be useful (both for knowledge and for management purposes) to understand if there are different responses of plant communities based on the different altitudinal vegetation belt.

This research aims to analyze the vegetation of two spruce forests of *Calamagrostio arundinaceae-Piceetum* damaged by the Vaia storm located in Lombardy (Camonica Valley) in the mountain belt and the high mountain belt, using two types of approach (remote sensing and floristic-ecological analysis) in order to enable a broad evaluation of the ecosystem responses to blowdown damage. In particular, the values and trends of the normalized difference vegetation index (NDVI) from 2018 (before the Vaia event) to the current situation (2021) were analyzed to evaluate the ability of the ecosystem to restore vegetation cover and greenness (density of green leaves). Moreover, floristic and ecological features of the current vegetation were analyzed in 2021 in order to develop models of plant succession trends useful to understanding the spontaneous processes in place and also for management goals.

## 2. Materials and Methods

### 2.1. Study Areas

Two study areas in the high Camonica Valley (Brescia Province, Lombardy region, Northern Italy), destroyed by the Vaia storm, were considered in this research: Val Malga (A) and Vione (B) (Figure 1).

Camonica Valley was one of the most affected areas, with about 800 ha of forests (mostly spruce forests) devastated, and 300,000 m^3^ of downed trees [21]. Study area A is in the municipality of Sonico (Latitude N: 46°07′34″; Longitude E: 10°24′54″) on a north-facing slope (~350°) at an elevation of 1550–1650 m a.s.l. (high mountain belt), covers about 8000 m^2^, and is within the boundaries of the Adamello Regional Park. Study area B is within the municipality of Vione (Latitude N: 46°14′49″; Longitude E: 10°25′53″) on a south-facing slope (~170°) at an elevation of 1200–1300 m a.s.l. (mountain belt), and this second area covers about 7000 m^2^. Both study areas belong to the Central and Eastern Alps ecoregional section (Alpine Province, Temperate Division) that includes a considerable amount of the Alpine Province, and is constituted by an oceanic temperate climate at the uppermost elevation and the temperate continental in interior valleys [22]. Upper Camonica Valley has a rainfall of more than 1000 mm per year, concentrated mainly in spring and autumn, while the annual average temperature is about 8.5 °C [9].

The two study areas are part of the acidophilic alpine series of silver fir and spruce (*Calamagrostio arundinaceae-Piceo excelsae sigmetum*) [23] in which the mature vegetation stage consists of coniferous forests (dominated by *Picea abies*) of *Calamagrostio arundinaceae-Piceetum* association (*Vaccinio-Abietenion* phytosociological sub-alliance, *Piceion excelsae* alliance, *Piceetalia excelsae* order, *Vaccinio-Piceetea* class) [23,24,25] (Appendix A). The two forests destroyed by the Vaia storm were *Calamagrostio arundinaceae-Piceetum* forests, which are the most widespread spruce forest association in this area of Lombardy [17,23], characterized by the following common plant species: *Picea abies* (dominant tree), *Larix decidua*, *Betula pendula*, *Sorbus aucuparia*, *Lonicera nigra*, *Calamagrostis arundinacea*, *Vaccinium myrtillus*, *Oxalis acetosella*, *Luzula nivea*, *Maianthemum bifolium*, and *Phegopteris connectilis*. From 2019 to 2021, a small part of the dead biomass of the two study areas was removed by the Alta Valle Camonica Forest Consortium.

### 2.2. Multispectral Images and NDVI Analysis

Four georeferenced multispectral images of the upper Val Camonica were acquired by the Copernicus Open Access Hub (https://scihub.copernicus.eu accessed on 1 September 2022). Each image was acquired by the Sentinel-2 satellite (satellite number: A) during the first half of July in 2018 (before Vaia), 2019, 2020, and 2021 (when the floristic analysis was carried out to analyze the current vegetation). Images with cloud cover < 10% and cloud shadow < 1% were chosen to reduce the errors in the calculation of the NDVI index. The pixel resolution of each image was 100 m^2^ (10 m × 10 m). The images were imported and analyzed using ArcGIS Desktop 10 software (Esri, Redlands, CA, USA). In detail, the “raster calculator” tool was used for the calculation of the NDVI index according to the following formula:(1)$$NDVI=\frac{NIR-RED}{NIR+RED}$$
where NIR and RED are the spectral reflectance in near-infrared and red bands, respectively. NDVI values range from −1.0 (areas with no or scarce vegetation) to 1.0 (areas with dense green vegetation).

The NDVI maps of each year of analysis were created to highlight the changes from 2018 to 2021. Moreover, the average NDVI value of the two study areas was calculated using the “summarize raster within” tool of ArcGIS to evaluate the NDVI trends from 2018 to 2021. The polygonal shapefiles of the two study areas affected by the Vaia storm were provided by the Comunità Montana di Valle Camonica, the public body that manages the forests of Camonica Valley.

### 2.3. Vegetation Sampling and Data Analysis

Forty phytosociological relevés were performed in the two sampling areas (21 in A and 19 in B) to gather information on the current plant communities 3 years after Vaia. In detail, the relevés were performed inside and outside (control) the forests damaged by the storm. Phytosociological relevés were performed in July 2021 according to the method of the Zurich–Montpellier Sigmatist School [26] and considering plots of 100 m^2^ (10 m × 10 m). The maximum height of the vegetation was measured in each relevé and plant species (tracheophytes) were determined using the “Flora d’Italia” dichotomous keys [27]. The cover index of each species was assigned using the conventional abundance-dominance scale of Braun-Blanquet: r, rare species in the relevé; +, coverage < 1%; 1, coverage 1–5%; 2, coverage > 5–25%; 3, coverage > 25–50%; 4, coverage > 50–75%; 5, coverage > 75–100%. The floristic data of the relevés were arranged in a matrix (relevés × species) where cover indexes were replaced into percentage values as proposed by Canullo et al. [28] (r, 0.01%; +, 0.50%; 1, 3.00%; 2, 15.00%; 3, 37.50%; 4, 62.50%; 5, 87.50%) in order to perform numerical and statistical analysis.

For each study area a hierarchical cluster analysis and detrended correspondence analysis (DCA) were performed to identify floristic similarities/differences among the relevés and highlight the main plant communities (clusters). Cluster analysis was performed using the unweighted pair group method with arithmetic mean method (UPGMA) and the chord distance coefficient [29]. The optimal number of clusters was assessed based on the “elbow” method that is based on the smallest number of clusters, which account for the largest amount of variation in the data. Plant species significantly associated with the different clusters (diagnostic species) were identified using Pearson’s phi coefficient (*Φ*) [30,31] as follows:(2)$$\Phi =\frac{N\cdot n_{p}-n\cdot N_{p}}{\sqrt{n\cdot N_{p}\cdot (N-n)\cdot (N-N_{p})}}$$
where *N* is the number of relevés in the dataset, *N_p_* is the number of the relevés in a target group of relevés (equalizing the size of relevés groups), *n* is the number of occurrences of the species in the dataset, and *n_p_* is the number of occurrences of the species in the target group of relevés. The values of *Φ* range from −1 to 1 (higher values indicate that species occurrences are concentrated in the target relevés group, whereas lower values indicate that they are under-represented in the target relevés group). Cluster analysis and DCA was performed using the “vegan” package [32] of R 3.6.1 software [33] while the diagnostic species identification was conducted using the “indicspecies” package [34]. The scientific names of the tracheophytes are in accordance with Pignatti [27].

Ecological and phytosociological analysis of the vegetation of the two study areas was performed in order to obtain a complete overview of the characteristics of the current vegetation and to hypothesize the models of plant succession. Plant diversity was estimated in terms of species richness, by counting the number of species recorded in each relevés, and the Shannon diversity index (HS) was calculated according to the following equation:(3)$$HS=-\sum_{i}^{N}p_{i}\ln p_{i}$$
where *p_i_* is the proportional abundance of the *i*-th species and *N* is the number of identified plant species. For the synecological analysis, the indices of Landolt et al. [35] (T, temperature; K, continentality; L, light intensity; F, soil moisture; R, substrate reaction; N, nutrients; H, humus; D, aeration) were applied while the hierarchical floristic classification system of vascular plant, bryophyte, lichen, and algal communities of the European vegetation [36], and other national and international resources [17,24,37,38,39], were consulted for phytosociological data. The names of the phytosociological classes are in accordance with Mucina et al. [36] while those of the lower *syntaxa* follow Andreis et al. [17] (for the groups of coniferous forests of Lombardy) and Chytrý et al. [39] (Appendix A). The presence of the identified alliances in the Southern Alps was verified using Preislerová et al. [40].

## 3. Results

### 3.1. NDVI Values and Trends

Figure 2 shows the NDVI maps of the two study areas (Val Malga, Figure 2a; Vione, Figure 2b) from July 2018 to July 2021. The maps show a drastic reduction in NDVI from July 2018 to July 2019 due to forest destruction by Vaia for both study areas (and other nearby disturbed forests). Instead, the comparison of the maps of 2019 and 2021 of each study area (Figure 2a,b) shows a slight increase in the shade of green, which suggests a slight increase in NDVI due to the partial restoration of vegetation cover/greenness. The increase in green hue is most noticeable in area B (Figure 2b), where there are more dark green patches within the area destroyed by the Vaia storm compared to area A (Figure 2a).

The increase in NDVI over time (from 2019 to 2021) is confirmed by the graph in Figure 3, which also shows how the impact of the storm was more destructive in area A. In fact, the two study forests had a very similar NDVI value (A: 0.790 ± 0.038; B: 0.801 ± 0.036) before the storm, while in July 2019 the value of that of area A was lower (0.511 ± 0.089) than that of area B (0.608 ± 0.158). Figure 3 shows a slightly faster rise in the NDVI for area B, which will reach a value like the one of 2019 in 2022 (4 years after the storm). The NDVI of A is likely to reach a similar value to pre-disturbance in 2024 (6 years after Vaia and 2 years later than area B).

### 3.2. Current Vegetation of Area A

A total of 92 species were identified in the vegetation study of area A (Appendix A). Most of them are common perennial species (herbs, shrubs, and trees) of the mountain areas of the Southern Alps. Figure 4 shows the dendrogram returned by the cluster analysis and the three different vegetation types (clusters) of area A:A1, *Picea abies* forest (*Calamagrostio arundinaceae-Piceetum*): vegetation present at the edges of the forest destroyed by the Vaia storm (control), consisting of spruce (*Picea abies*), which is the dominant tree, and other diagnostic species of the *Vaccinio-Piceetea* class, (Table 1) including *Larix decidua*, *Vaccinium myrtillus*, *Arctostaphylos uva-ursi*, and Oxalis acetosella.A2, *Rubetum idaei*: shrubland dominated by *Rubus idaeus*, which is a diagnostic species of this cluster together with *Luzula sylvatica* (Table 1). In this community, there are also herbaceous species of forest edges (such as *Epilobium angustifolium*) and young woody plants such as *Sorbus aucuparia*, *Sambucus racemosa*, *Salix caprea*, and *Abies alba* (silver fir). Juvenile spruces are scarce. *Rubetum idaei* is a secondary vegetation type of clearing, forests opening, and degraded forest. The height of this vegetation is generally less than 1.5 m.A3, *Piceo abietis-Sorbetum aucupariae*: community with Sorbus aucuparia (growing mostly as a shrub) accompanied by other woody plants and shrubs of the *Sambuco-Salicion capreae* alliance (*Robinietea* class) as *Salix caprea* (diagnostic species of this cluster), *Sambucus racemosa*, *Betula pendula*, and *Rubus idaeus*. *Calamagrostis villosa* and *Abies alba*, two species that characterize the silver fir woods of Lombardy (*Calamagrostio villosae-Abietetum*), also grow in this plant community. This is a short-living community occurring on forest clearings, where it is gradually replaced by the vegetation of mature forests. Vegetation height: 1–3.5 m.

Figure 5 shows the DCA biplots of the relevés––therefore, the dynamic relationships between the various vegetation clusters. Along the first axis (DCA1), the species that characterize the spruce forests (A1), such as *Picea abies* and *Vaccinium myrtillus*, abruptly decrease. Instead, from A2 to A3, the coverage of *Rubus idaeus* gradually decreases while that of the trees of *Piceo abietis-Sorbetum aucupariae* (such as *Salix caprea*) increases.

The ecological features of the three vegetation types are reported in Table 2. Cluster A2 and A3 and differ from A1 (control) by having higher values of T, L, R and N, and lower values of species richness, HS, K, and H. A3 differs from A2 by having fewer species and higher values of HS, L, and D.

### 3.3. Current Vegetation of Area B

In study area B, 92 species were identified (Appendix A). Most of them are common in the hilly and mountain areas of Lombardy Alps. Four types of vegetation were identified (Figure 6):

B1, *Picea abies* forest (*Calamagrostio arundinaceae-Piceetum*): this vegetation is physiognomically similar to A1 but with some species of the mountain-hilly belt, typical of broad-leaved woods, such as *Quercus petraea* (sessile oak), *Luzula nivea*, *Veronica officinalis*, *Campanula rotundifolia*, *Corylus avellana*, and *Prunus avium* (Table 1 and Appendix A). This is the vegetation present at the edges of the forest destroyed by Vaia (control).B2, community of *Astrantio-Corylion avellanae* alliance: shrubland with species of the *Crataego-Prunetea* class such as *Corylus avellana* (diagnostic species of this cluster), *Rosa canina*, and *Prunus spinosa*, and other pioneer shrubs and trees such as *Populus tremula*, *Salix caprea*, *Buddleja davidii* (exotic species), *Betula pendula*, and *Sambucus nigra*. Spruce is scarce while *Quercus petraea* is fairly abundant. The height of the trees/shrubs is about 1.5–3.0 m.B3, *Brachypodium rupestre-Astragalus glycyphyllos* community: grassland dominated by *Bachypodium rupestre* with other herbaceous diagnostic species of this cluster (*Achillea millefolium*, *Lotus corniculatus*, *Astragalus glycyphyllos* and *Festuca valesiaca*) and young shrubs (*Corylus avellana*, *Rosa canina*, *Rubus idaeus*, *Salix caprea* and *Prunus avium*). In this community, there are larches over 10 m high that survived Vaia.B4, *Rubetum idaei*: plant community dominated by *Rubus idaeus*, unique diagnostic species of this cluster (Table 1). In this plant community, there are also young shrubs/trees (maximum plant height: 1.5 m) such as *Corylus avellana*, *Salix caprea*, *Betula pendula*, *Fraxinus excelsior*, and *Quercus petraea*. No young/juvenile spruce was detected.

Figure 7 shows the DCA biplots of the relevés of area B. Along the first axis (DCA1), the communities B2, B3, and B4 are clearly separated by the spruce forest (B1). In fact, along the DCA1, the species of *Vaccinio-Piceetea* (as *Picea abies*) increase while the shrubs, *Rubus idaeus*, decrease. Along the DCA2 axis, on the other hand, shrubs (*Corylus avellana* in particular) and young broad-leaved trees decrease, while the coverage of *Brachypodium rupestre* increases (Figure 7). The results of the ecological analysis (Table 2) show marked differences between the spruce forest (B1) and the other plant communities. In particular, B1 has the lowest values of species richness, HS, T, L, and R. Table 2 also shows a difference between the soil properties of communities B2 and B3. In particular, B3 requires less humid, more aerated, and poorer soil nutrients than B2.

### 3.4. Plant Succession Models of the Two Study Areas

Figure 8 shows two models of plant succession trend (one for each study area) developed following the NDVI analysis and the floristic-ecological study of the current vegetation. In both models, the post-Vaia communities have different ecological characteristics (radar charts) than those of pre-Vaia forests (spruce forests), especially regarding the climatic indicators T and L. In fact, in both models the current vegetation has more thermophilic and heliophilous species in comparison with spruce forests. Based on the ecological and phytosociological characteristics of the current vegetation and the young woody species identified, the two trees that could become dominant in the mature forests of the two study areas (excluding anthropogenic interventions) are the following: *Abies alba* in area A and *Quercus petraea* in area B (Figure 8). The values of the ecological indexes of Landolt (radar charts in Figure 8) of these two species are more similar to those of current vegetation than to those of pre-Vaia forests.

## 4. Discussion

Our results show similar ecological processes upon Vaia in the two study areas. NDVI analysis showed how the first stages of the two secondary successions are inducing an increase in the green plant cover and/or density. The current vegetation analysis, though, did show that the presence of young trees typical of mature forest communities could develop completely different forests (from a floristic, physiognomic and ecologic point of view) from the spruce forests prior Vaia.

We can see how the NDVI is increasing in both areas, but at a different speed (faster in B). The pre-Vaia value (2018) is, in fact, very similar for the two areas (~0.8) while it is markedly different in 2019 and the following years, which is in line with the evaluation of the NDVI values (Figure 3). This aspect could be due to two main factors: the different intensity of the storm disturbance and the different elevation (with all that this entails) in the two areas. A different wind speed and storm intensity is included in the predictors of forest vulnerability together with forest characteristics, forest management history, and soil, site, and, of course, climate conditions [41], with the latter (climate conditions), in our case, linked to the location of the site (different elevation and vegetation belt, and different exposition), the consequences of which will be further discussed.

Regarding the intensity of the storm, it is known that as the intensity increases, the NDVI of the damaged area decreases [42]. If we consider the different damage in the two areas, it is possible to see how in area A, all the tall trees were downed by the storm, while in area B, a few larches and spruces survived despite being broken/damaged by the wind (Figure 1). In fact, the variation of NDVI from 2018 to 2019 is −0.28 for area A and −0.19 for area B. These data are similar to those obtained by [43] for forests severely (−0.25) and moderately (−0.18) damaged by fire.

Lesser damage to tall tree vegetation in area B could justify a faster increase in NDVI from 2019 to 2022, since vegetation recovered quicker from the event (tall trees and shrubs expanded the canopy sooner). Moreover, area B is situated at an altitude of about 1250 m a.s.l. (300 m lower than area A) and is south facing. This would guarantee to area B a longer (about 1 month) and milder vegetation period compared with area A. In the Alpine area, in fact, we can estimate a growing season shorter by 1 week for every 100 m of elevation increase [44,45]. The different values (and trends) of NDVI in the two study areas could also be due to the different water stress (and health status) of the plants. In fact, the NDVI is one of the widely used indices for measuring water stress [46]. In the two study areas, any differences in the water stress of the plants could be due to the different quantities and frequencies of rainfall, the different water capacity of the soil, and the different resistance to water stress of the different plant species. These aspects should be investigated by carrying out climatological, pedological/hydrological, physiological, and/or ecological studies.

NDVI was useful to map the forests affected by wind blowdown (Figure 2) and to monitor the changes in vegetation cover/density over time (Figure 2 and Figure 3). From the maps in Figure 2, we can see clearly in 2019 “patches” with low NDVI that were absent in 2018. These patches overlap with the study areas and other forests damaged by Vaia coherently with the data gathered by the foresters of the “Comunità Montana di Valle Camonica” territorial body. The periodic comparison of NDVI values of a specific geographic area or the comparison of NDVI value before and after a disruptive climatic event could help define this kind of blowdown damage beside evidencing changes in the land-use [47].

NDVI applied in this way in our study did not give “qualitative” information, meaning typology and/or floristic composition of the plant communities, on the vegetation of the two areas. This does not allow an evaluation of the maturity of the forests and/or the stage of plant successions. NDVI average values of area B in 2018 and 2021 were very similar (~0.8) (Figure 3), though there were differences in the vegetation. Furthermore, the model in Figure 3 shows how the NDVI values pre-disturbance could be reached in 4–6 years. In this short time-span, though, the formation of a mature forest community is highly improbable. In the opinion of Giupponi et al. [9], indeed, forest reconstitution in mountain areas in Camonica Valley would need at least 20–25 years if the area was favored by opportune strategies of ecological restoration, and much more (more than 60 years) in the case of spontaneous regeneration [48].

Although the NDVI data in this study could be affected by some errors due to the different activity of the vegetation in relation to the climatic factors of the various years considered [49], it is plausible that in a few years (less than 10) in the two study areas, there could be high NDVI values without the presence of mature forest vegetation (the final stage of the plant succession). This would happen due to the presence of the dense canopy constituted by shrubs (*Rubus* spp., *Sambucus* spp. and *Corylus avellana*) and pioneer species (*Betula pendula*, *Salix caprea*, *Populus tremula*, and *Sorbus aucuparia*), which does not belong to the final stage of the succession. To evaluate whether the model with which the NDVI trend was estimated (Figure 3) is correct, it would be good to calculate the NDVI of the two study areas over the next 5–20 years by applying methods/tools that can minimize the errors due to technical limitations [50,51]. In fact, according to Valtonen et al. [52], the NDVI values of areas affected by forest restoration increase only in the first 10 years of forest regeneration and then stabilize (from 10 to 25 years) around 0.8. The results of the research of Sun et al. [50] also showed a “greening” phase in the 10 years that followed the environmental restoration works of a river corridor in China. This “greening” phase, if it is also be observed for the regeneration of forests in the Alps (such as those under study), could be used as an indicator of the success of forest recovery [52] in combination with other ecological indices (based on the floristic features of the vegetation) developed to evaluate the success of environmental restoration works [53].

Despite the existence of recent applications of NDVI for the definition of forest plant communities [54], the most used method to date is the field data collection through phytosociological relevés. This phytosociological method enables an understanding of the processes governing vegetation assembly and dynamics [55].

*Rubetum idaei* was found both in the high mountain belt (area A) and in the mountain belt (area B) by the current vegetation analysis, where it is, nonetheless, associated with more thermophilic species (Table 2). This kind of vegetation has a low species richness and low HS due to the presence of a dominant species (*Rubus idaeus*), which is an indicator of forests subject to recent disturbances [56]. *Rubetum idaei* will be gradually replaced by other plant communities when the canopy of trees and shrubs will grow higher than about one meter. In this way, *Rubus idaeus* (a markedly heliophilous pioneer species) would miss the direct light necessary for it to thrive.

The shrubs and young trees with this role are those of *Piceo abietis-Sorbetum aucupariae* (in area A) and *Astrantio-Corylion avellanae* (in area B), representing the next (intermediate) stage of the two successions (Figure 8). To date, the shrublands mentioned above can be found in those patches of the two study areas where Vaia was less disruptive and/or other environmental factors (such as soil conditions, seeds bank, already present young trees, microclimate, etc.) favored the species of the intermediate stages in competition with *Rubus idaeus*. A similar situation could occur for the *Brachypodium rupestre-Astragalus glycyphyllos* community, which is the analogue of *Rubetum idaei* in area B, where soil is poorer and drier.

The intermediate stages show some species of the pioneer stages as *Rubus idaeus* (although with low plant cover) together with some relict species of spruce forest (Figure 5 and Figure 7), but also young trees characterizing mature forests different from the spruce wood (*Calamagrostio arundinaceae-Piceetum*). In area A, in fact, we could find *Abies alba*, which in Lombardy characterizes woods of *Calamagrostio villosae-Abietetum* [23,24]; while in area B, we found *Quercus petraea*, which characterizes the acidophilous oak forests of the hilly and sub-mountain belts of the Southern Alps [23,24,25,57]. Both these species are more thermophilic compared to spruce, especially *Quercus petraea* [35]. Sessile oak and silver fir are generally found at lower elevation compared to the spruce forests in the Italian Alps [25,58].

Radar charts in Figure 8 showed how the current vegetation of the study areas has ecological needs more like *Abies alba* (area A) and *Quercus petraea* (area B) than the *Picea abies* forests pre-Vaia, especially considering the temperature factor (T index). If the trend remains unvaried, the mature forests in the two areas could develop in more thermophilic ecosystems, with *Abies alba* and *Quercus petraea* replacing partially or totally spruce ones. This shift could be attributed to the forest management changes rather than climatic disturbances. Inhabitants favored the expansion of the spruce forest in the Italian Alps for the higher commercial value [25,58]. For this reason, mountain and high mountain silver fir woods and sub-mountain oak woods in Camonica Valley were replaced with the spruce forest when possible. Today, nevertheless, the abandonment and the lack of management of the forests in those areas allowed the comeback of *Abies alba* and *Quercus petraea*.

*Abies alba* and *Quercus petraea* forests expansion upwards could be further favored by milder temperatures at higher elevation due to climate change. There is a tendency of low-altitude species to migrate to higher elevations in altitudinal belts of mountains [59,60]. Lenoir et al. [59] observed a significant upward shift in forest plant species in western Europe. In more detail, the authors detected an upward shift in species optimum elevation averaging 29 m per decade, and that the shift is larger for species restricted to mountain habitats and for grassy species characterized by a faster population turnover. The plant species upward shift determines a concomitant upward shift of altitudinal vegetation belts [61,62]. Considering Camonica Valley, our research detected a coherent effect, since today *Quercus petraea* woods can grow up to 1300 m a.s.l., which means 300 m higher compared with the Italian vegetational series map [23,63]. Conifers could therefore find their ecological optimum over 1300 m (unless there is a special exception), where *Abies alba* should then prevail on *Picea abies*.

The elevation shifts of forest species and vegetation belts as suggested by this research should be investigated further in Camonica Valley and other Southern Alps areas in general. Furthermore, detailed studies are needed to understand the responses of plants (and plant communities) of the southern Alps to the main effects of severe storms, such as canopy aperture. When most of the canopy remains undamaged, there will be little space for other sources, and, consequently, few changes in community features, while substantial canopy damage opens the way for important effects from the other sources [64].

Severe storms are natural disturbances that alter the dynamics of succession by creating canopy gaps (of various sizes), and the most visible impact is the increase in light in the undergrowth. Light is one of the most limiting resources for plant growth. The rapid disclosure of the overstorey creates a limited opportunity for increased rates of regrowth and growth in the understorey, especially for light-demanding pioneer species [65]. The results of research conducted on a subtropical forest [66] revealed that the increase in light in the understorey, caused by the canopy trimming, stimulated the germination from the seed bank and significantly increased both the recruitment of pioneer plants and their density. The same research found that a canopy opening interval and its influence on pioneer tree recruitment is the dominant factor driving short-term recovery and can alter long-term forest structure and composition. Studies similar to those conducted by Shiels et al. [66] should be also be conducted in the forests of the southern Alps (including boreal-like conifer forests) to understand if there are similar ecological responses to those of subtropical forests.

Another aspect to consider would be the major or minor resistance of the future forest plant communities to disturbances such as windstorms. In our case, for example, the future silver fir and sessile oak forests can be expected to be more resilient to windthrow compared to spruce, since the latter has a more superficial root apparatus [67,68]. Differences among forest plants in rooting depth, above-ground architecture, and wood strength can cause considerable variations in the probability of damage [69,70] beyond the effects of size, since, of course, older and higher trees are subjected to more windthrow [64].

As well as forest trees, other components of the ecosystem considered will likely undergo important changes, such as, for example, the entomofauna, and the consequences of this playing again on the vegetation shifts. The increased amount of dead wood in the forests was certainly an important result of the disturbances caused by Vaia, despite the prompt and valuable actions of the collection and re-use of downed trees by the land managers. It has already been mentioned that some forest pests such as the spruce bark beetle (*Ips typographus*, *Coleoptera*: *Scolytidae*) reproduce in newly wind-downed trees, and the number and dimension of recently downed spruces were found to be the most important features influencing the colonization of this pest [71]. Windstorms such as Vaia could provide a surplus of breeding sites with many recently downed trees, and this disturbance frequently elicits an increase in the attacks at living spruce trees, too [72]. Another aspect to consider is the attractiveness of the host volatiles emitted by fresh spruce wood [73], which is obviously higher in the case of large patches of recently fallen trees, and which act in synergy with the aggregation pheromones of bark beetles [74]. In conventional situations, these beetles kill isolated spruces in distress conditions, such as, for example, forest gaps [75]. They very rarely attack viable trees due to the difficulty of defeating the defensive systems of a healthy spruce tree [76]. In the case of an event of windstorm, the downed trees show low or no defenses, and beetles can colonize them more easily. They then increase in number and become dangerous for healthy trees, too.

An interesting development of this study would be to analyze what is happening in the patches attacked by bark beetles, since this pest is typical of spruce forest and changes in the plant communities could also affect the resilience to some pests such as *Ips typographus*. This effect could be reinforced by the mechanism of plant-insect communication, since non-host volatiles appear to disturb the reaction of bark beetles to pheromones [77], meaning that mixed mature forests could be less preferable as sites of bark beetle attack, and that those attacks might even be prevented.

Other positive changes due to the Vaia action could also be the increase in the habitat for many innocuous species, such as, for example, several rare species of *Cerambycidae* or other endangered insects [78]. Saproxylic beetles play an important role in decomposition processes and the consequent nutrient cycling in natural ecosystems, and many are also involved in the pollination of many alpine plants.

Furthermore, the development of plant communities such as the *Rubetum idaei* after the disturbance event, where important foraging resources for solitary and domestic bees as *Rubus idaeus* are present [79], could favor the communities of pollinators in higher mountain habitats, providing an environment richer in foraging resources compared with the previous *Calamagrostio arundinaceae-Piceetum* spruce forests. Additionally, the clearings in the canopy create more nesting sites for some solitary bees such as bumblebees, too [80,81]. There is certainly a need for further studies to help land managers in maintaining a balance between forestry management and biodiversity conservation.

## 5. Conclusions

This research has enabled, for the first time, an understanding of what the responses of the vegetation after the Vaia storm are in two damaged forests of *Calamagrostio arundinaceae-Piceetum* in Lombardy (Southern Alps). The two forests considered, although in two different vegetation belts (mountain and high mountain), are undergoing similar ecological processes. The NDVI values of both drastically decreased in 2019 (because of Vaia) and gradually increased in the following years (from 2019 to 2021). We estimated a return to the NDVI values pre-Vaia in less than 10 years, even if our model is validated and/or improved by applying more accurate NDVI analysis methods. Nevertheless, the floristic and ecological analysis of the current vegetation allowed us to hypothesize the plant succession trends of the two study areas.

In the two forests, a secondary plant succession is underway, characterized by the primary stage of *Rubetum idaei*, to which different shrubs follow (intermediate stages), where the presence of young trees of the *Quercus petraea* and the *Abies alba* species are indicative of mature forest communities different from the original pre-Vaia spruce forest (for the floristic, physiognomic, and ecologic resiliency to windstorm features). We hypothesize that a mixed forest of silver fir and spruce (*Calamagrostio villosae-Abietetum*) will form in the area located in the high mountain belt, and a sessile oak forest in the study area located in the mountain belt. Although the succession models presented in this research must be confirmed by diachronic studies of the vegetation, the presence of young trees typical of more thermophilic mature forest communities than the pre-disturbance ones demonstrates an upward shift of forest plant species in Camonica Valley, probably due to the different forest management in the last decades in addition to climate change. This phenomenon could lead to significant changes in the mountain ecosystems of Val Camonica (and the Southern Alps), which will have to be the subject of extensive ecological studies that will be useful for territorial stakeholders to protect and valorize the forests.

## Figures and Tables

**Figure 1 plants-12-01369-f001:**
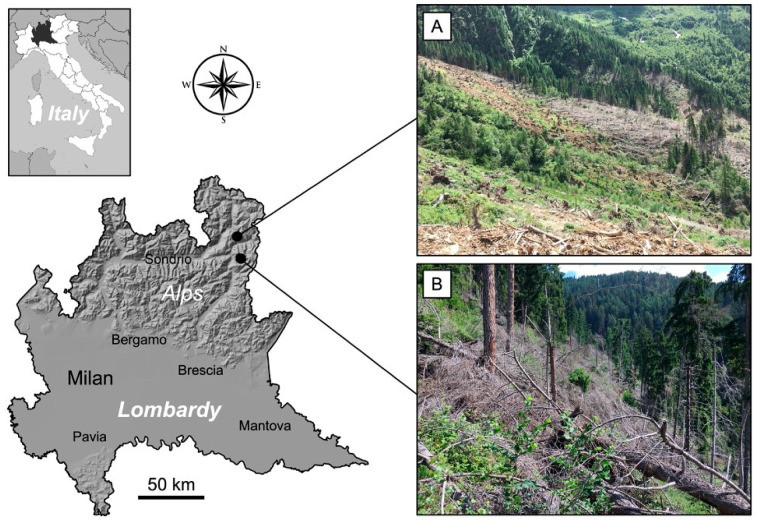
Location of the two study areas: Val Malga (**A**) and Vione (**B**). The photographs show the areas in July 2021.

**Figure 2 plants-12-01369-f002:**
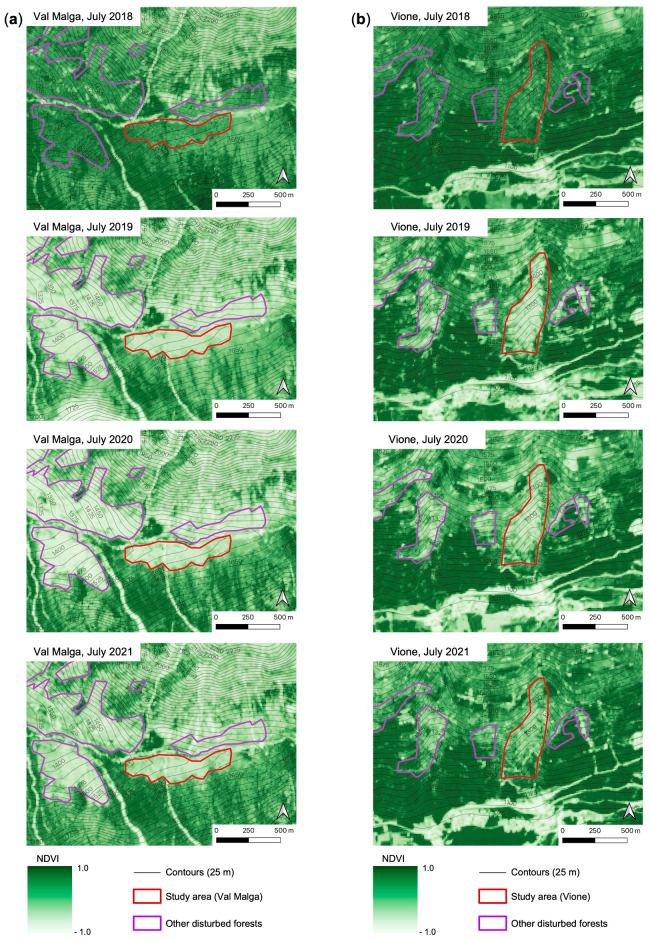
NDVI maps of Val Malga (**a**) (high mountain belt) and Vione (**b**) (mountain belt) in July 2018 (before Vaia storm), 2019, 2020, and 2021. The borders of the two study areas are in red while the purple lines delimit other forests disturbed by the Vaia storm (data provided by Comunità Montana di Valle Camonica).

**Figure 3 plants-12-01369-f003:**
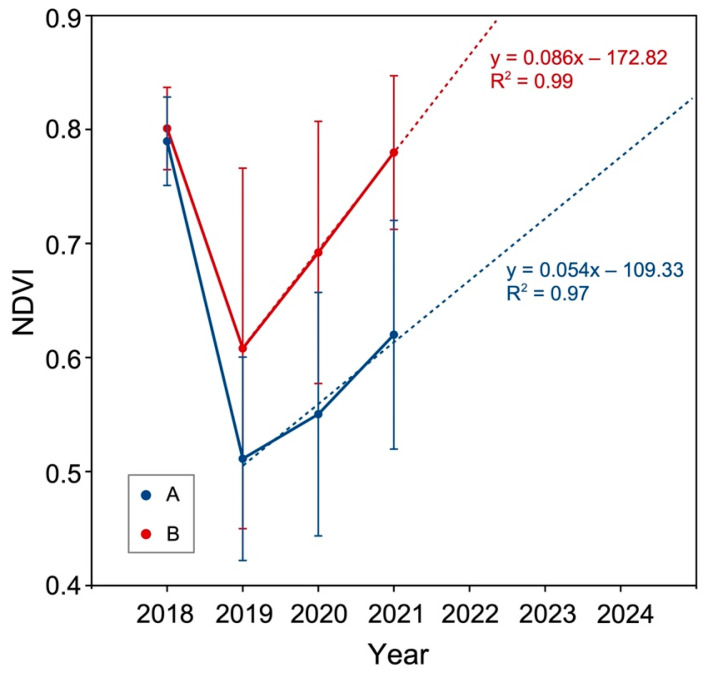
Average value (dot) ± standard deviation (bar) of NDVI of the two study areas (A, Val Malga; B, Vione) from July 2018 (before Vaia storm) to July 2021. The broken lines indicate the trends of the NDVI after the Vaia storm.

**Figure 4 plants-12-01369-f004:**
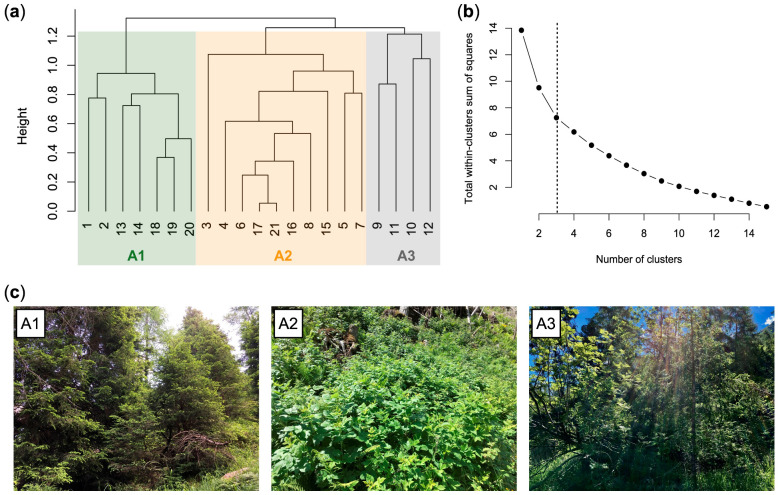
Dendrogram of the relevés of study area A (**a**), distribution of total within-clusters sum of squares by number of relevé groups distinguished via hierarchical clustering (**b**), and photographs of the three vegetation types (**c**). Key: A1, *Picea abies* forest; A2, *Rubetum idaei*; A3, *Piceo abietis-Sorbetum aucupariae*.

**Figure 5 plants-12-01369-f005:**
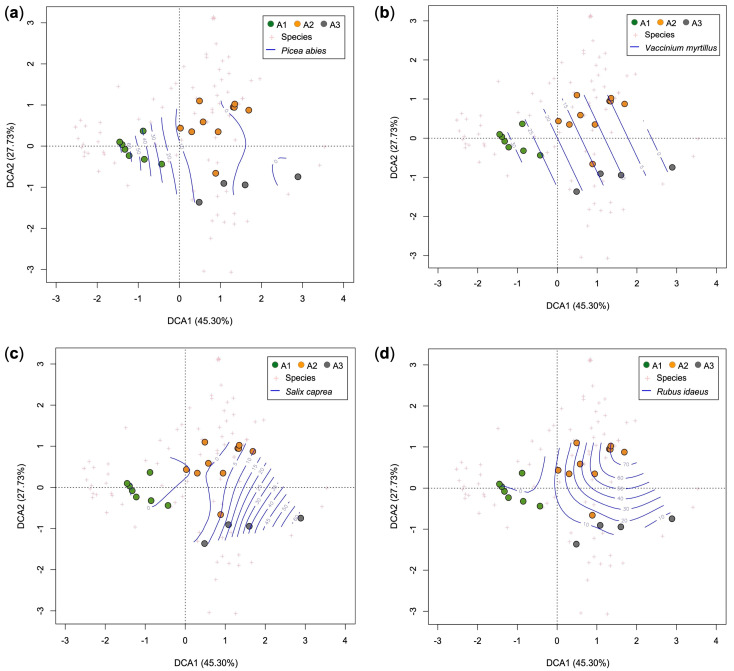
DCA biplots of phytosociological relevés (dots) overlapped with plant cover contour lines (blue lines with percentage of plant cover) of some diagnostic species of each cluster: *Picea abies* (**a**), *Vaccinium myrtillus* (**b**), *Salix caprea* (**c**) and *Rubus idaeus* (**d**). Key: A1, *Picea abies* forest; A2, *Rubetum idaei*; A3, *Piceo abietis-Sorbetum aucupariae*.

**Figure 6 plants-12-01369-f006:**
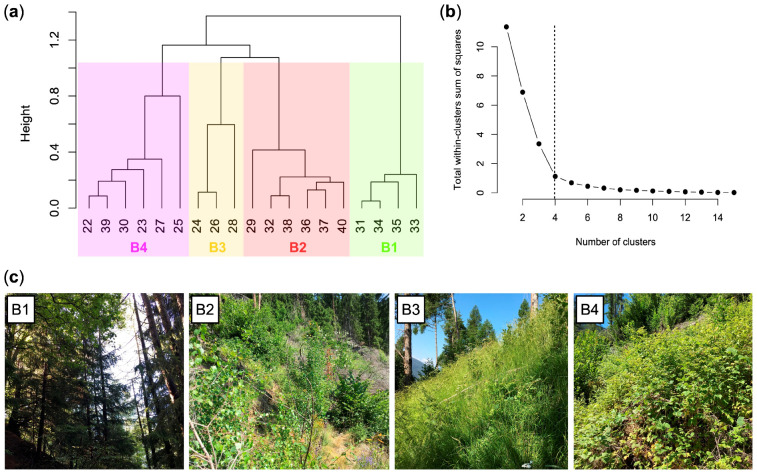
Dendrogram of the relevés of study area B (**a**), distribution of total within-clusters sum of squares by number of relevé groups distinguished via hierarchical clustering (**b**), and photographs of the four vegetation types (**c**). Key: B1, *Picea abies* forest; B2, *Astrantio-Corylion avellanae*; B3, *Brachypodium rupestre-Astragalus glycyphyllos* community; B4, *Rubetum idaei*.

**Figure 7 plants-12-01369-f007:**
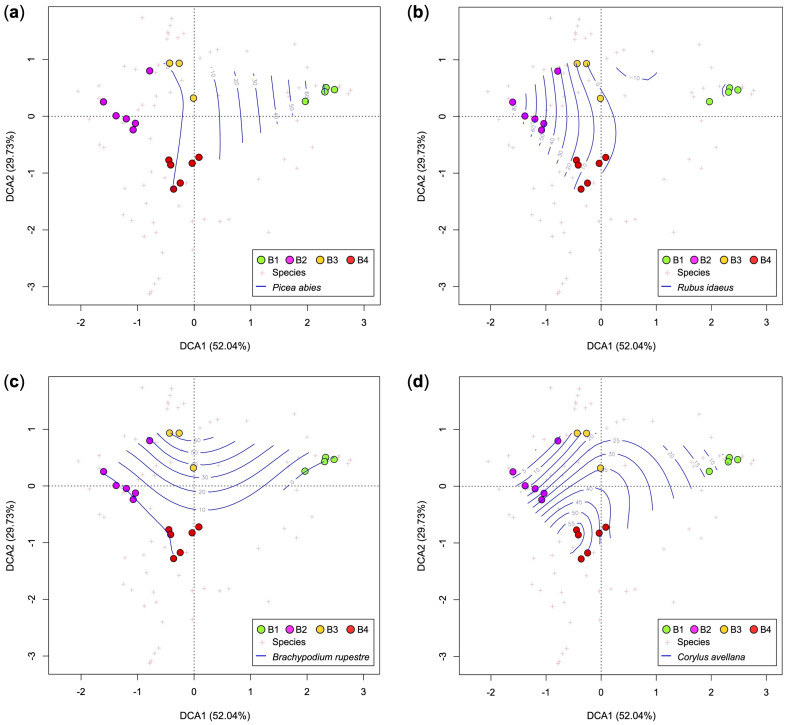
DCA biplots of phytosociological relevés (dots) overlapped with plant cover contour lines (blue lines with percentage of plant cover) of some diagnostic species of each cluster: *Picea abies* (**a**), *Rubus idaeus* (**b**), *Brachypodium rupestre* (**c**) and *Corylus avellana* (**d**). Key: B1, *Picea abies* forest; B2, *Astrantio-Corylion avellanae*; B3, *Brachypodium rupestre-Astragalus glycyphyllos* community; B4, *Rubetum idaei*.

**Figure 8 plants-12-01369-f008:**
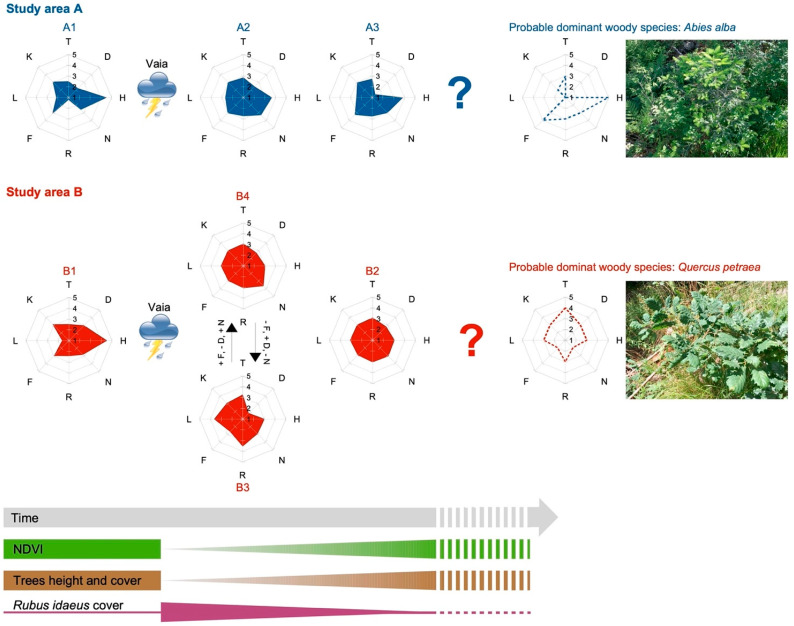
Models of plant succession trends of the two study areas (A, Val Malga; B, Vione). The models show the radar charts of the ecological indices of Landolt et al. [35] of each plant community and the schematic trend of NDVI, tree height and cover, and *Rubus idaeus* cover. The values of the ecological indexes of the two probable dominant woody species (broken line in the radar chart) of the mature forest community and a photograph of them in the two study areas are reported. Key of the ecological indexes: T, temperature; K, continentality; L, light intensity; F, soil moisture; R, substrate reaction; N, nutrients; H, humus; D, aeration. Key of plant communities: A1, *Picea abies* forest; A2, *Rubetum idaei*; A3, *Piceo abietis-Sorbetum aucupariae*; B1, *Picea abies* forest; B2, *Astrantio-Corylion avellanae*; B3, *Brachypodium rupestre-Astragalus glycyphyllos* community; B4, *Rubetum idaei*.

**Table 1 plants-12-01369-t001:** Diagnostic species of each plant community (cluster). The codes of the clusters are the same ones used in the dendrograms. Phytosociological class code, Pearson’s phi coefficient (*Φ*), and *p*-value of each diagnostic species are reported. The codes of phytosociological class are the same used in [36]: PIC, *Vaccinio-Piceetea*; LOI, *Loiseleurio procumbentis-Vaccinietea*; QUE, *Quercetea robori-petraeae*; NAR, *Nardetea strictae*; FAG, *Carpino-Fagetea sylvaticae*; VIR, *Betulo carpaticae-Alnetea viridis*; MUL, *Mulgedio-Aconitetea*; ASP, *Asplenietea trichomanis*; MON, *Montio-Cardaminetea*; ROB, *Robinietea*; EPI, *Epilobietea angustifolii*; RHA, *Crataego-Prunetea*; ERI, *Erico-Pinetea*; FES, *Festuco-Brometea*; MOL, *Molinio-Arrhenatheretea*; ART, *Artemisietea vulgaris*; GER, *Trifolio-Geranietea sanguinei*. Key: *, significant (*p* < 0.05); **, significant (*p* < 0.01).

Study Area	Cluster	Species	Phytosociological Class	*Φ*	*p*	
A	A1	*Picea abies*	PIC	0.89	0.01	**
		*Larix decidua*	PIC	0.74	0.01	**
		*Vaccinium myrtillus*	PIC, LOI, QUE, NAR	0.70	0.01	**
		*Arctostaphylos uva-ursi*	PIC, LOI,	0.69	0.01	**
		*Oxalis acetosella*	PIC, FAG	0.65	0.01	**
		*Viola biflora*	VIR, MUL, MON, ASP	0.55	0.03	*
		*Rhododendron ferrugineum*	LOI	0.54	0.02	*
		*Imperatoria ostruthium*	MUL	0.47	0.05	*
	A2	*Rubus idaeus*	ROB	0.66	0.01	**
		*Luzula sylvatica*	PIC, FAG, QUE,	0.56	0.05	*
	A3	*Salix caprea*	ROB	0.61	0.01	**
		*Lonicera alpigena*	FAG	0.53	0.04	*
		*Petasites albus*	MUL, EPI	0.45	0.04	*
B	B1	*Picea abies*	PIC	0.98	0.01	**
		*Luzula nivea*	FAG	0.72	0.03	*
		*Veronica officinalis*	QUE, NAR	0.68	0.02	*
	B2	*Corylus avellana*	RHA	0.79	0.01	**
	B3	*Brachypodium rupestre*	ERI, FES	0.92	0.01	**
		*Lotus corniculatus*	MOL	0.90	0.01	**
		*Achillea millefolium*	MOL, ART	0.83	0.01	**
		*Acer pseudoplatanus*	FAG	0.78	0.01	**
		*Astragalus glycyphyllos*	GER	0.69	0.01	**
		*Festuca valesiaca*	FES	0.63	0.05	*
		*Fraxinus excelsior*	FAG	0.58	0.04	*
	B4	*Rubus idaeus*	ROB	0.93	0.01	**

**Table 2 plants-12-01369-t002:** Ecological features of the vegetation types of the two study areas (A, Val Malga; B, Vione): A1, *Picea abies* forest; A2, *Rubetum idaei*; A3, *Piceo abietis-Sorbetum aucupariae*; B1, *Picea abies* forest (of area B); B2, *Astrantio-Corylion avellanae*; B3, *Brachypodium rupestre-Astragalus glycyphyllos* community; B4, *Rubetum idaei* (of area B). Key: HS, Shannon diversity index; T, temperature; K, continentality; L, light intensity; F, soil moisture; R, substrate reaction; N, nutrients; H, humus; D, aeration.

Study Area	Vegetation Type	Richness	HS	T [1-5]	K [1-5]	L [1-5]	F [1-5]	R [1-5]	N [1-5]	H [1-5]	D [1-5]
A	A1	25.14 ± 6.67	1.90 ± 0.09	2.50 ± 0.13	3.07 ± 0.17	2.14 ± 0.25	3.11 ± 0.07	1.20 ± 0.22	2.60 ± 0.18	4.54 ± 0.29	2.21 ± 0.31
	A2	23.30 ± 7.93	1.72 ± 0.79	2.85 ± 0.12	3.01 ± 0.12	2.68 ± 0.27	3.05 ± 0.11	2.73 ± 0.23	3.32 ± 0.40	3.65 ± 0.50	2.46 ± 0.38
	A3	19.25 ± 0.96	1.81 ± 0.61	2.74 ± 0.22	2.96 ± 0.03	2.46 ± 0.14	3.27 ± 0.19	2.75 ± 0.29	3.02 ± 0.39	3.86 ± 0.68	1.41 ± 0.22
B	B1	15.00 ± 3.83	0.90 ± 0.28	2.52 ± 0.06	3.14 ± 0.08	1.62 ± 0.30	2.97 ± 0.03	2.41 ± 0.16	2.83 ± 0.08	4.54 ± 0.25	2.98 ± 0.02
	B2	26.67 ± 6.71	1.54 ± 0.46	3.10 ± 0.05	3.02 ± 0.04	3.09 ± 0.09	2.94 ± 0.05	3.03 ± 0.02	3.09 ± 0.17	3.04 ± 0.08	2.79 ± 0.10
	B3	26.00 ± 2.64	1.56 ± 0.25	3.27 ± 0.17	3.10 ± 0.07	3.66 ± 0.14	2.66 ± 0.11	3.54 ± 0.26	2.95 ± 0.06	3.02 ± 0.02	1.78 ± 0.34
	B4	19.17 ± 5.56	1.17 ± 0.41	3.07 ± 0.09	3.02 ± 0.02	3.09 ± 0.18	2.97 ± 0.08	3.06 ± 0.16	3.65 ± 0.22	3.03 ± 0.02	2.72 ± 0.32

## Data Availability

The datasets used and/or analyzed during the current study are available from the corresponding author upon reasonable request.

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
