# Peer review of "Restoration of Vegetation Greenness and Possible Changes in Mature Forest Communities in Two Forests Damaged by the Vaia Storm in Northern Italy"

_plants, 2023, doi:10.3390/plants12061369_

Round 1

Reviewer 1 Report

Comment

This manuscript is interesting because it deals with monitoring of two forests damaged by the Vaia storm and explains a gradual restoration of vegetation greenness in mature forest communities. I review this manuscript and provide following comment for authors.

1.      Title uses “but” that is not suitable. I suggested that authors could consider to change title, such as “A gradual restoration of vegetation greenness changes in mature forest communities and using two forests damaged by the Vaia storm in Northern Italy as an illustration”.

2.      Abstract is suitable for this paper. I only have a slight suggestion. Because the time of study periods is not so long, using global warming to infer in the end of Abstract might be not suitable.

3.      As international audiences, they may not so understand the Vaia storm or the relationship between the Vaia storm and forests. I suggest that it should have more description in Introduction or Discussion chapter.  

4.      This study used two plantations as an illustration to analysis a gradual restoration of vegetation greenness changes in mature forest communities. I could understand the representative of these two plantations. If possible, more descriptions of background information in Introduction or Materials and Methods is necessary.

5.      Each equation should add number.

6.      In Figure 2, the difference of A and B should have more explanation in text.

7.      The results are significant and could provide information for fields of plant ecology

8.      In Discussion chapter, I suggest that author should emphasize the mechanism of the Vaia storm and forests. If possible, adding more references to discuss it is necessary.

Over all, this manuscript is interesting and I recommend it for publication in Plants after minor revision.

Author Response

  1. Title uses “but” that is not suitable. I suggested that authors could consider to change title, such as “A gradual restoration of vegetation greenness changes in mature forest communities and using two forests damaged by the Vaia storm in Northern Italy as an illustration”.

Authors response: We modified the title following your suggestion to make it more descriptive of the content of the paper. See main text.

  1. Abstract is suitable for this paper. I only have a slight suggestion. Because the time of study periods is not so long, using global warming to infer in the end of Abstract might be not suitable.

Authors response: thank you for your very reasonable observation. We modified the last sentence of the abstract by removing "global warming" (see the main text, line 24-26). The new sentence is also more suitable for the special issue "Responses of Plants and Plant Communities to Environmental Changes in Mountain Areas" and more descriptive of the outcomes of the paper.

  1. As international audiences, they may not so understand the Vaia storm or the relationship between the Vaia storm and forests. I suggest that it should have more description in Introduction or Discussion chapter.

Authors response: Thank you for your suggestion. We added some details about Storm Vaia (and the area where the study forests are located) at the end of the introduction: lines 76-80.

  1. This study used two plantations as an illustration to analysis a gradual restoration of vegetation greenness changes in mature forest communities. I could understand the representative of these two plantations. If possible, more descriptions of background information in Introduction or Materials and Methods is necessary.

Authors response: We considered forests A and B because in the two study areas there were forests of Calamagrostio arundinaceae-Piceetum, one of the most widespread spruce forest association in Lombardy. Furthermore, these forests were in two different vegetation belts (A: high mountain belt; B: mountain belt), so it was interesting to observe whether the restoration of the vegetation was different. We have added these details in the introduction (84-91) as well as in materials and methods (122-129).

  1. Each equation should add number.

Authors response: we added numbers to each equation.

  1. In Figure 2, the difference of A and B should have more explanation in text.

Authors response: We added further explanations in the text (lines 215-223) (and in the caption of figure2) regarding the differences between the two study areas.

  1. The results are significant and could provide information for fields of plant ecology.

Authors response: we agree that the results of this research will be useful in the field of plant ecology. In fact, we think that the phenomenon of upward shift of plant species needs detailed ecological studies that could help to understand the changes in the mountain ecosystems of Val Camonica (and of the southern Alps more generally). We emphasised this aspect in the conclusions. In the discussions, we also added that NDVI could be used with ecological indices (based on the study of vegetation) to evaluate the success of forest restoration.

  1. In Discussion chapter, I suggest that author should emphasize the mechanism of the Vaia storm and forests. If possible, adding more references to discuss it is necessary.

Authors response:  In the discussions we have given emphasis to the consequences of severe storms on forests by adding even more bibliographic references. See lines 392-396, 401-403, 512-532, 536-539.

Over all, this manuscript is interesting and I recommend it for publication in Plants after minor revision.

Authors response:Thank you for your positive feedback and your recommendations to improve the paper.

Reviewer 2 Report

Using the normalized difference vegetation index (NDVI) from 2018 to 2021, this paper analyzed the values and trends of NDVI and evaluated the ability of the ecosystem to restore vegetation cover and greenness in forests located in Camonica Valley (Northern Italy, Southern Alps). This paper may contribute to further understand the spontaneous processes in place and for management goals. However, there are some concerns that the authors should address before it can be considered for publication.

1. Why did the authors only used the NDVI data from 2018-2021 for analysis. 

2. In study areas, the author mentioned that two research areas of high Camonica Valley have similar climate, location and vegetation type. What is the important implication of studying the impact of storms on these two regions? 

3. In order to further highlight the innovation of this article, it is better to compare the results of this study with other studies.

4. More mechanism explanations should be added to further explain the different impacts of storms on NDVI in the two areas.

5. A paragraph of limitation discussion should be added to clarify the limitation or uncertainty of data and methods in this current study. For example, there may be some uncertainties of this study's results due to the inaccuracy of NDVI data (Rao et al., 2015; Shen et al., 2022).

6. The conclusion is not a simple restatement of the results. The authors should further clarify the contribution of the research results to the research field.

7. To be consistent with the other results in this study, I suggest keep two decimals for all the values in the Table 1.

References:

An improved method for producing high spatial-resolution NDVI time series datasets with multi-temporal MODIS NDVI data and Landsat TM/ETM+ images. Remote Sensing, 2015, 7(6), 7865-7891.

Vegetation greening, extended growing seasons, and temperature feedbacks in warming temperate grasslands of China. Journal of Climate, 2022, 35, 5103-5117.

Author Response

  1. Why did the authors only used the NDVI data from 2018-2021 for analysis.

Authors response: we used NDVI data from 2018 to 2021 to evaluate the NDVI trend considering the period between the situation immediately before Vaia and the year in which the floristic-vegetation analysis was carried out (2021). We specified in the text (in the introduction and in materials and methods) that the current situation corresponds to the year in which the phytosociological relevés were carried out (2021), see lines 94 and 138-140.

  1. In study areas, the author mentioned that two research areas of high Camonica Valley have similar climate, location and vegetation type. What is the important implication of studying the impact of storms on these two regions? 

Authors response: although the two study areas have similar forest vegetation (Calamagrostio arundinaceae-Piceetum), they are located in two different altitudinal vegetational belts: A, high montane belt; B, montane belt. We specified this detail in the introduction (lines 84-91) as well as in materials and methods (lines 122-129).

  1. In order to further highlight the innovation of this article, it is better to compare the results of this study with other studies.

Authors response: We added several comparisons between our results and those in literature. See chapter Discussion.

  1. More mechanism explanations should be added to further explain the different impacts of storms on NDVI in the two areas.

Authors response: We added further discussions of the different NDVI values of the two study areas. See the main text (lines 410-417).

  1. A paragraph of limitation discussion should be added to clarify the limitation or uncertainty of data and methods in this current study. For example, there may be some uncertainties of this study's results due to the inaccuracy of NDVI data (Rao et al., 2015; Shen et al., 2022).

Authors response: We added a paragraph explaining that our NDVI data may be inaccurate, and we added that in the next few years the NDVI trend of the two study areas could be monitored using more accurate methods. Lines: 437-441 and 444-455

  1. The conclusion is not a simple restatement of the results. The authors should further clarify the contribution of the research results to the research field.

Authors response: We modified the conclusions as suggested. See main text.

  1. To be consistent with the other results in this study, I suggest keep two decimals for all the values in the Table 1.

Authors response: done.

Round 2

Reviewer 2 Report

The authors have addressed all my concerns. I suggest accept this paper in its present form.